# Global terrestrial invasions: Where naturalised birds, mammals, and plants might spread next and what affects this process

Henry Häkkinen ¤*, Dave Hodgson, Regan Early *

Centre for Ecology and Conservation, Faculty of Environment, Science and Economy, University of Exeter, Penryn, United Kingdom

¤ Current address: Institute of Zoology, Zoological Society of London, London, United Kingdom
* henry.hakkinen@ioz.ac.uk (HH); r.early@exeter.ac.uk (RE)

## Abstract

More species live outside their native range than at any point in human history. Yet, there is little understanding of the geographic regions that will be threatened if these species continue to spread, nor of whether they will spread. We predict the world's terrestrial regions to which 833 naturalised plants, birds, and mammals are most imminently likely to spread, and investigate what factors have hastened or slowed their spread to date. There is huge potential for further spread of naturalised birds in North America, mammals in Eastern Europe, and plants in North America, Eastern Europe, and Australia. Introduction history, dispersal, and the spatial distribution of suitable areas are more important predictors of species spread than traits corresponding to habitat usage or biotic interactions. Natural dispersal has driven spread in birds more than in plants. Whether these taxa continue to spread more widely depends partially on connectivity of suitable environments. Plants show the clearest invasion lag, and the putative importance of human transportation indicates opportunities to slow their spread. Despite strong predictive effects, questions remain, particularly why so many birds in North America do not occupy climatically suitable areas close to their existing ranges.

## Introduction

Understanding and predicting the spread of introduced species is one of the key conservation and ecological challenges of the 21st century [1]. However, we know little about what causes the introduced range of some species to increase rapidly, while other species remain in small, isolated populations years after establishing self-sustaining populations [2,3]. This major gap in our understanding prevents us from understanding how much of invasive spread is due to characteristics of the invader or the invaded environment. The most imminent threat is posed by the many thousands of species that are naturalised outside their native range and may continue to spread much more widely. However, there is a surprising lack of attention paid to the potential spread of already-naturalised species. Without understanding what has affected the spread of these species historically, we can assess neither which species are likely to spread

**Data Availability Statement:** Some of the occurrence and trait data used throughout this study cannot be directly shared, however the data can be downloaded from the original online

repositories. All the relevant code, data we are allowed to share, and links to download data we cannot share directly are available from: https://doi.org/10.5281/zenodo.8205905. We have linked to this in the data accessibility section. Some intermediate data we created are too large to place in a repository (shapefiles of each species estimated native and naturalised ranges, rasters of niche filling and expansion, diagnostic and validation statistics), and so are available upon request from the authors. Nonetheless, the data provided and linked to in the zenodo repository permits full replication of our final results. Where we cannot include original data, we have created a dummy dataset to allow researchers to run and explore our methods.

**Funding:** HH was funded by the Natural Environment Research Council Great Western Four + Doctoral Training Partnership (NERC GW4+ DTP) studentship program (Grant Number 102681). https://www.nercgw4plus.ac.uk/ The funders had no role in study design, data collection and analysis, decision to publish, or preparation of the manuscript.

**Competing interests:** The authors have declared that no competing interests exist.

**Abbreviations:** ALA, Atlas of Living Australia; DIC, deviance information criterion; LOO, leave-one-out; MCMC, Markov chain Monte Carlo; PCA, principal components analysis; SDM, species distribution model; TSS, true skill statistic; WAIC, widely applicable information criterion.

further nor the geographic regions that will be most affected. This hampers pro-active management, of both already-introduced species and those yet to be introduced.

Species' potential naturalised ranges are often calculated by "climate matching" to the conditions occupied in the native range [4]. This approach is widely used for invasion risk assessment [4]; however, the portion of this potential range that naturalised species fill is extremely variable [5, 6]. Some species spread rapidly [7, 8], and may even expand beyond native climate conditions [9], while others remain restricted to a portion of their potential range long after any expected invasion lag [6, 10]. The spatial distribution of suitable environmental conditions, introduction history, and species characteristics, all influence naturalised ranges [11], but their relative importance is unknown.

Concerning the spatial distribution of environmental conditions, climate matching rarely, if ever, accounts for the connectivity of suitable climate. In a landscape where suitable climate is continuous, it should be much easier for a species to spread than in a landscape with small and isolated fragments of suitable climate [12]. Concerning introduction history, time since introduction is a key factor predicting naturalised species range size [13–15]. The number of individuals introduced ("propagule pressure") also has an important role, especially soon after establishment [16,17]. Species introduced into habitat that is similar to their native range are more likely to establish and spread [18]. Species dispersed by humans are frequently more successful at spreading widely [14,19,20]. Concerning species characteristics, species that are able to increase population size quickly, compete effectively, adapt to novel environments, and spread widely are more likely to spread widely [21,22].

Evaluating the relative contribution of the above factors to range filling is crucial for invasion management and the approach best suited to predicting spread. For example, if low range filling is predominantly driven by invasion time lags and dispersal limitation, then already-introduced species could spread much more widely than they have to date [6]. In this case, we should focus research on spatial population dynamics. If species' naturalised ranges are limited by habitat fragmentation [23], then we can categorise risk between landscapes and manage landscapes appropriately.

We undertake the first global assessment of the degree to which introduced birds, mammals, and plants, ranging from small annual herbs to long-lived large trees and from tiny herbivores to apex predators, have occupied climatically suitable areas in their introduced ranges. Our study is the first, to our knowledge, to incorporate climatic suitability when investigating invasion lags, and accounting for this major determinant of species' naturalised ranges frees us to examine other drivers of spread robustly. We ask whether introduction history, species traits associated with establishment and spread, and the spatial distribution of suitable areas have hastened or slowed the spread of species. We identify geographic regions with striking invasion deficits, where many already-naturalised species could spread much more widely. We examine whether our evidence suggests that these species are indeed likely to continue to spread.

## Results

### Summary of introduced species and range filling

Every continent on Earth (except Antarctica) hosts at least 1 native and naturalised species from our dataset (S1 Fig). In their native ranges, birds occupied a median of 397 grid-cells, mammals 240, and plants 471 cells. Using range polygons, birds had a median native range size of 1.8 million km$^2$, mammals 1.7 million km$^2$, and plants 2.0 million km$^2$. In all 3 taxonomic groups, naturalised range sizes were smaller: median values are 112,000 km$^2$ for birds, 283,000 km$^2$ for mammals, and 441,000 km$^2$ for plants (Table 1).

**Table 1. Correlates of range filling for each taxonomic group and model verification.**

| | Model parameter | Estimate | 95% CI | Differences between realms? | Model verification | Estimate |
|---|---|---|---|---|---|---|
| **Plants** | Intercept | −2.30 | (−1.68, −2.99) | Aus | Sample size | 484 |
| | Years since introduction | 0.35 | (0.53, 0.16) | | DIC | −1,065.79 |
| | Days till flowering (logged) | −0.13 | (0.02, −0.26) | Aus, Nea | pD | 18.96 |
| | Local recording effort | −0.30 | (0.29, −0.88) | Aus | Pseudo R-squared | 0.31 |
| **Birds** | Intercept | −2.79 | (−2.12, −3.41) | | Sample size | 50 |
| | Years since introduction | 0.35 | (0.76, −0.02) | Nea | DIC | −190.10 |
| | Dispersal distance (km) | 0.37 | (0.72, 0.05) | | pD | 19.00 |
| | Fragmentation (contagion) | 0.61 | (1.11, 0.07) | | Pseudo R-squared | 0.59 |
| **Mammals** | Intercept | −1.83 | (−1.15, −2.56) | | Sample size | 46 |
| | Dispersal distance (logged km) | −0.28 | (0.07, −0.65) | | DIC | −87.95 |
| | Fragmentation (contagion) | 0.64 | (1.10, 0.19) | | pD | 22.20 |
| | | | | | Pseudo R-squared | 0.50 |

The proportion of climatically suitable areas successfully occupied by introduced species was in general low, but highly variable across species (S1 Table). Mammals occupied a median of 4% of available range, birds 1%, and plants 5%.

The areas to which the greatest number of already-naturalised birds could spread are in Mexico, the south-eastern and the south-western USA. Eastern Europe could face the spread of the greatest number of naturalised mammals. The greatest number of naturalised plant species could spread to the southern and eastern USA (and small pockets in north-western USA), Mexico, Eastern Europe, and southern, central, and western Australia (Fig 1). The geographic results come with the caveat that risk estimates can be affected by recording bias, the role of which we explore further below.

## Introduction history, species characteristics, and climate connectivity affecting range filling

Models successfully converged after 20,000 iterations. The proportion of range filling was highly variable across realms (Table 1 and S1 Table), and range filling was particularly high in Australia for plants and mammals, though not for birds (Table 1). For plants, time since introduction significantly increased range filling globally, age at first flowering significantly decreased range filling in Australia and the Nearctic, recording effort significantly decreased range filling in Australia and the Nearctic (Table 1 and Figs 2 and S2). For birds, contagion of suitable climate and natal dispersal distance significantly increased range filling globally, and the number of years since introduction significantly increased range filling in the Nearctic (Table 1 and Figs 2 and S3). For mammals, contagion of suitable climate significantly increased range filling globally. Natal dispersal distance was retained in the final model, even after several rounds of model simplification (i.e., removing it reduced model fit), though it did not have a statistically "significant" effect globally or in any realm (Table 1 and Figs 2 and S4).

## Validation and sensitivity analyses

We identified areas where low recording effort could affect the number of species present or with potential to spread (S5 and S6 Figs).

We estimated the accuracy of climate matching predictions using the true skill statistic (TSS) scores generated using cross-validation. The median TSS score across all species was 0.62 (standard deviation = 0.21), which is considered to be "substantial" performance [24].

**Plants**

**Birds**

**Mammals**

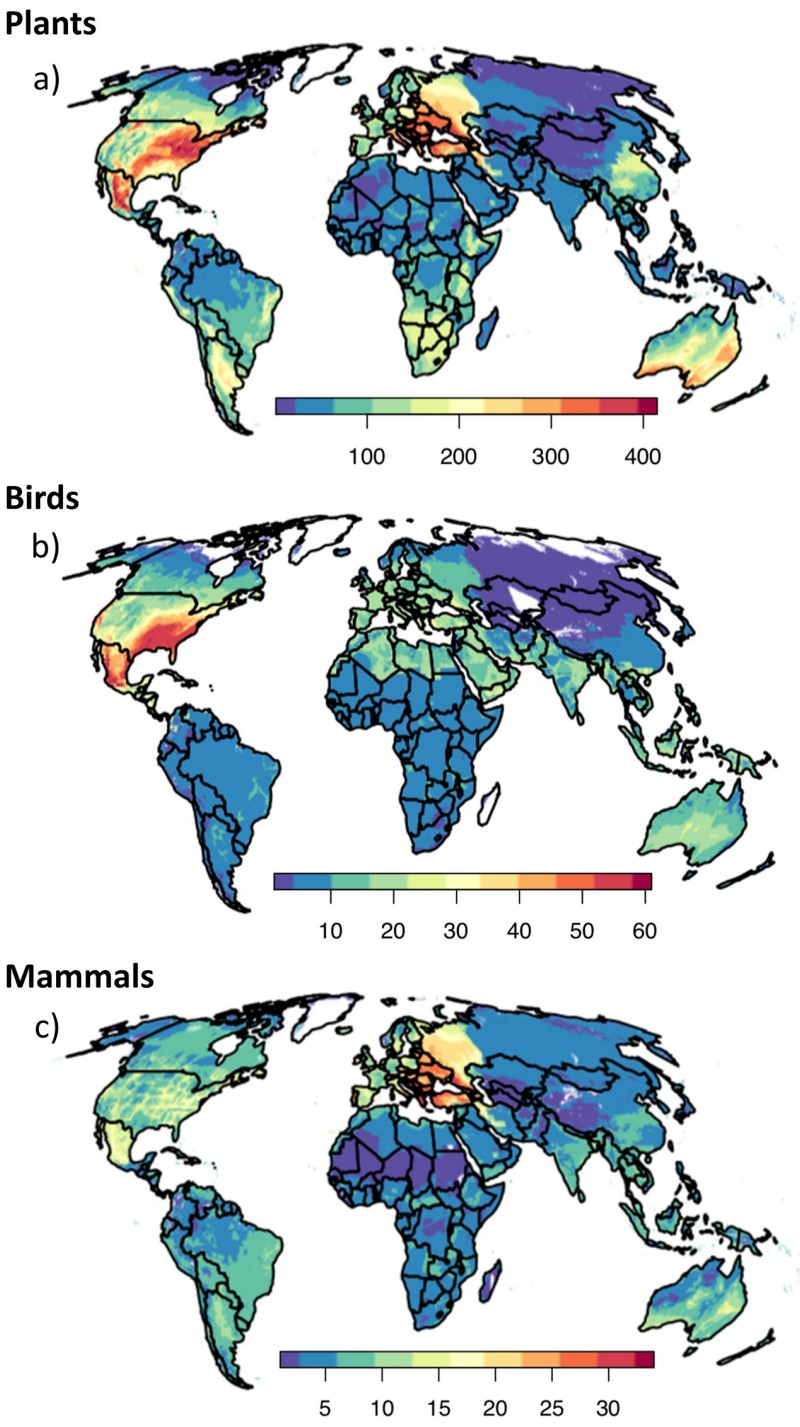

**Fig 1. Areas where regionally naturalised species could expand further.** The numbers of (a) plants, (b) birds, and (c) mammals that could spread to each 10-min grid-cell based on the cell's climatic suitability for each species. The data underlying this figure can be found in https://doi.org/10.5281/zenodo.8205905. Country and continent outlines were produced by the International Working Group on Taxonomic Databases for Plant Sciences (TDWG), specifically the WGSRPD Level 4 boundaries; data and usage notes can be found at (https://github.com/tdwg/wgsrpd).

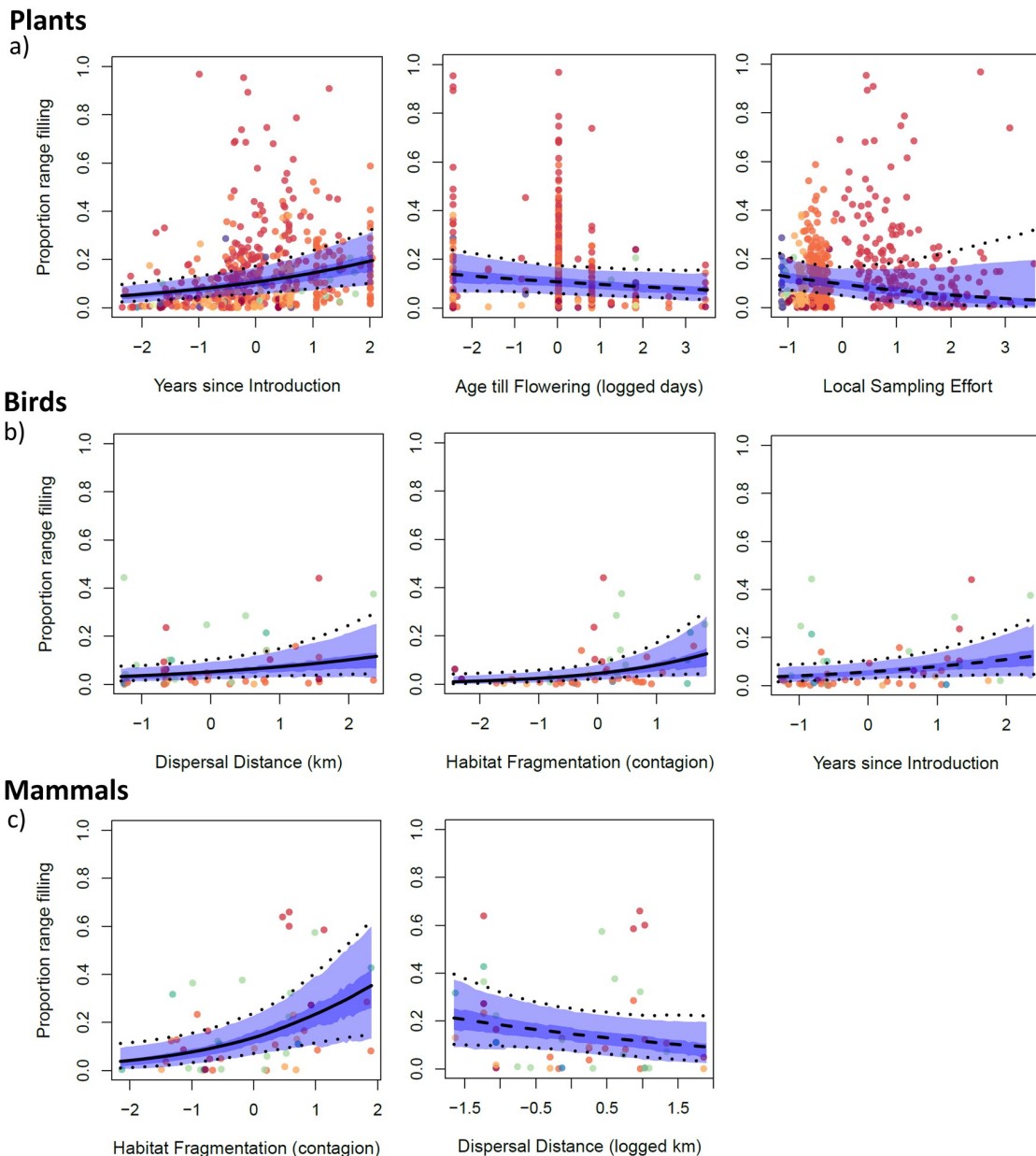

**Fig 2. Global parameter estimates of variables retained in the final Bayesian hierarchical model to correlate with species range filling for (a) plants, (b) birds, and (c) mammals.** A solid line signifies the estimate was consistently above or below 0 in >95% of simulations (and therefore judged as significant), a dashed line means it was not. The lighter shaded area shows the 95% probability density interval for the parameter estimate, and the darker shows the 50% interval. For regional effects, see Supporting information (S2–S4 Figs). The data underlying this figure can be found in https://doi.org/10.5281/zenodo.8205905.

Species' native presences in validation data were predicted very accurately; the median proportion of presences predicted to be in suitable climate was 0.99 (standard deviation = 0.06) and only 2 species had a prediction rate of under 0.6. Thus, it is unlikely that we underpredicted species' potential naturalised ranges, based on the native range data available.

Analyses of range filling seem unlikely to have been affected by inclusion of species that have been recorded as naturalised in few grid-cells. Species range filling did increase when

species have naturalised in more grid-cells, but the association was very weak for species that have naturalised in few grid-cells (S7 Fig). In a separate analysis, we did find that species niche filling increased with the number of grid-cells in which a species had naturalised, but with a very high degree of variability (S8 Fig). Therefore, to be sure that the number of grid-cells occupied did not affect estimates of what affects range filling, we re-ran the final model of factors affecting range filling for plants that had naturalised in >20 grid-cells. This analysis produced similar parameter estimates to the final model for species that had naturalised in >5 grid-cells (S2 Table), so the threshold of 5 was kept for all analyses.

When restricting potential ranges to a smaller portion of the native climate niche, geographic patterns of threat remain broadly similar (S9 Fig). However, the potential for birds to spread in North Africa and the Arabian Peninsula, and the potential for mammals to spread in Eastern Europe, does decline relative to other regions. The absolute number of plants that could spread within grid-cells in the Nearctic, Eastern Europe and Australasia, birds in the Nearctic, and mammals in Eastern Europe, declines by 40% to 50%.

Estimates for parameters retained in the final model are given as the mean estimate of all posterior draws, with the 2.5% and 97.5% estimates as confidence intervals in parentheses. Parameter estimates are given as the linear slope of the logit link equation. When parameter estimates vary across realms, this is indicated by providing the names of the realms in which it varies (Aus = Australian, Nea = Nearctic, Neo = Neotropical). Model verification data are given for the final models, including sample size, deviance information criterion (DIC) of the model, the effective number of parameters (pD), and correlation of the linear predictor against the link transformed response given as a pseudo R-squared. Note that a negative effect of recording effort means that more recorder effort in the potential naturalised range corresponded to lower range filling. See also S13 Table.

## Discussion

The world is in no way saturated with naturalised species, even if no new species become naturalised in the future. Nearly all species we studied have yet to expand throughout most of the areas that are climatically suitable for them, within the biogeographic regions where they have naturalised. This is despite substantial time to invade: 25% have been established for over 150 years. The potential spread of naturalised species appears greatest in regions that are already heavily invaded, i.e., North America, Australia, and Europe (Fig 1). However, already-naturalised species also threaten areas of the world thought of as less invaded (Fig 1). In South America and Southern Africa around 200 regionally naturalised plants have the potential to spread widely—a number comparable to that in Australia and China, which have historically borne the brunt of biological invasions [25–27]. Nonetheless, the potential for regionally naturalised species to spread in sub-Saharan Africa and the north of South America (Fig 1) is lower than one would expect from the globally high numbers of species naturalised there [1]. This is potentially encouraging given the recent increases in regional trade and transportation infrastructure in these regions [28] that could increasingly facilitate the spread of naturalised species. The potential for regionally naturalised species to spread in the eastern USA (particularly birds) and Mexico (particularly birds and plants) is higher than one would expect from the relative numbers of species naturalised there (Fig 1) [1]. Differences might be influenced by the broader taxonomic range measured in Dawson and colleagues [1] than that for which we could conduct climate matching. Nonetheless, the contrast illustrates how invasion threat cannot be characterised solely by the number of species naturalised.

The global potential for spread is high because range filling was consistently low (S1 Table). Why are such large areas as-yet unreached by species that have naturalised elsewhere in the biogeographic realm, and what does this suggest for the risk of spread?

Of the 3 taxonomic groups, time since introduction seemed to limit plant range filling most strongly. We suggest this is because plants generally have low dispersal [29] and are therefore most prone to time lags. However, dispersal ability itself did not significantly limit plants, potentially because long-distance dispersal events are so rare that they have little effect on range filling over the decades or centuries since introduction. Plant spread is often strongly dependent on human-assisted dispersal [19, 20]. Therefore, time since introduction may increase range filling by allowing for greater human transportation rather than natural dispersal. We note that horticulture does not seem to be the predominant human transportation mechanism here, as species associated with horticulture did not have significantly higher range filling than non-horticultural species. The potential importance of human transportation suggests that the threat in Fig 1a could be averted. For example, hundreds of plant species are naturalised in California (S1 Fig) and separated from climatically suitable areas in the eastern USA by mountains and desert. Only a few naturalised species have been transported between western and eastern USA [30], so controls on transport could be effective. Plants were the only taxon in which a non-dispersal trait affected range filling: flowering at an earlier age increased range filling, suggesting that a quick reproductive cycle assists range spread more so than natural dispersal over the decades or centuries since introduction.

Like plants, bird range filling was affected by time since introduction, but dispersal ability increased and contagion of climatically suitable areas decreased range filling. For mammals, while time since introduction limited range filling slightly when analysed alone, it was not retained in the final model. As with birds, contagion decreased mammal range filling, though there was no effect in the Nearctic, the realm where the potential for mammal spread was highest (Figs 1, 2, and S4).

Dispersal affected mammal range filling, but in the opposite direction to that for birds: strong mammalian dispersers tended to fill less of their potential naturalised ranges than weak dispersers. Mammal dispersal ability is highly positively correlated with body size [31], which typically correlates with having a larger home range size, requiring larger amounts of habitat [32]. In a preliminary analysis, both body mass and dispersal ability were trialled, but the high co-variance meant both could not be retained and of the 2, dispersal ability was retained as it was a better predictor. Therefore, difficulty establishing robust populations may hinder the spread of large-bodied species, masking any positive effect of dispersal ability. This interpretation is supported by the negative effect of contagion on mammal range filling, as fragmented habitat could slow both population establishment and natural dispersal. The contrasting results for dispersal and contagion between plants and the better dispersing mammals and birds suggest that fragmentation of suitable areas is more important when natural dispersal contributes heavily to spread.

We found no effect of population growth, generalism, or behavioural flexibility traits for mammals and birds, despite all having been shown to correspond to invasion success elsewhere [16, 17, 33]. It seems that for the relatively long-dispersing birds and mammals, once the availability of suitable climate is accounted for, interactions with native species and population growth rate are less important than simply being able to arrive in a new location.

We note that despite having the longest dispersal abilities in our dataset, birds displayed the lowest range filling. This may be because bird species had large native niches that cover a wide range of climates, a trend also found in multi-taxonomic comparisons of thermal niche width [34], and therefore have larger potential ranges than other taxa (particularly in the Afrotropical and Neotropical realms, S1 Table).

Both mammals and plants had strong differences in range filling between realms, with the highest level of range filling in Australia (S1 Table). This may be because of strong human influences on naturalised species ranges in Australia [35], where species have been introduced at or near multiple coastal cities throughout the areas climatically suitable for many of the species. Human-assisted dispersal from several population centres could help species spread, particularly plants [36]. Additionally, the dissimilar ecological assemblage between Australia and most other realms in the world could cause low biotic resistance [37]. In this case, a lack of ecologically similar species may reduce competition and predation and promote population growth and spread in the introduced species [38–40].

Although pseudo-$R^2$ values are convincing (Table 1), there remains unexplained variation in range filling. Of particular interest is the 26 birds introduced to Florida, of which 10 have not spread into any nearby climatically suitable areas in south-eastern USA, despite extensive transportation infrastructure and lack of apparent barriers (Fig 1). This is particularly surprising given high dispersal ability of birds. Some of these bird species may be restricted to their introduction locations because they are not truly self-sustaining and may be instead be supplemented by ongoing accidental releases [41]. Nevertheless, the large number of bird species that have not spread beyond Florida is still surprising. Potentially important factors we did not investigate directly are propagule pressure [14, 16, 17], repeated introductions that boost population size and genetic heterogeneity [16, 17], and the climatic suitability of initial sites of introduction [18]. In addition, introduced individuals often originate in a single part of the species' native range. These individuals may have narrower climate tolerances than those of the species as a whole [42], and may only ever be able to fill a small part of the potential range calculated using the entire native range. If such local adaptation is behind the widespread lack of range unfilling, climate matching could be highly prone to overpredict potential ranges. While birds introduced into Florida show the most striking example of unexplained range filling, the above arguments could apply to other taxa and locations, and thus climate-matching should be treated with caution.

Variation in recorder effort [43] could affect our results in 3 ways. First, in understudied regions we may under-record species' naturalised ranges (S5B, S5D, and S5E Fig) and thus overestimate the area to which they have yet to spread. However, the only effect of local recording effort we saw was the opposite: low effort corresponded with a slight increase range filling for plants (Table 1), an effect that was only significant in the Australian realm.

Second, the numbers of naturalised species may be underrepresented in under-recorded regions. It is likely our dataset overrepresents the number of naturalised species in the Nearctic, Palearctic, and Australasia (S10 Fig). Analyses exploring the possible effect of recording effort (S6 Fig) suggests that with less biased estimation of species numbers, eastern USA might no longer stand out as a global hotspot for the spread of naturalised birds (though the threat would remain moderate) and the threat in Mexico might become relatively low. The threat of bird spread in North Africa, the Arabian Peninsula, western Russia, south and south-eastern Asia, which is moderate based on raw species numbers (Fig 1), could have been substantially underestimated, and potentially be greater than the threat in eastern USA. Underestimation of naturalised species numbers could have obscured a large area of threat in China and pockets in Brazil. Underestimation would not alter the high threat from mammals and plants in parts of Eastern Europe and western Russia, but could have over-emphasised the threat in Western Europe relative to less well-recorded regions. One reason for geographic bias in species numbers is that we discarded species for which very little data were available. Many of these may be species that have narrow native ranges and climate niches, and have not spread widely once introduced, so pose relatively low threat. Therefore, adjusting for recorder effort does not necessarily represent threat more accurately than using uncorrected data. It is entirely plausible

that the species in our dataset have greater potential naturalised ranges and ability to spread than those we excluded.

Third, if the native geographic ranges of naturalised species are under-recorded (S5A, S5C, and S5E Fig), we might underestimate species' potential naturalised range size. This is unlikely to be the case for mammals and plants, since the native ranges of most species studied include the well-studied Western Europe and North America (S1A, S1C, and S1E Fig). However, many of the naturalised birds we studied are native to south Asia, South America, and sub-Saharan Africa, which have relatively low recording effort, so estimates of bird potential spread may be more conservative than those for mammals or plants. Moreover, there is now considerable evidence that many naturalised species undergo niche expansion, spreading to areas outside their native climate niche [9, 44–46]. Therefore, our predictions of all naturalised species' potential ranges could be conservative. We note there is no evidence of geographic trends in niche expansion, so this should not affect relative regional threat [9].

Given the potentially important effects of recorder effort, we suggest all large-scale threat assessments should explicitly explore the consequences.

A notable artefact in our results is the sharp division in invasion potential along the realm division between the east and west Palearctic, along the line of the Urals. This is a result of our realm definitions, as we did not consider species naturalised in one realm able to spread to the adjacent realm. Without this division, species native to East Asia would be considered native to Western Europe, and vice versa, when in reality many species have been introduced between the 2 regions by humans over recent centuries. Consequently, the number of species that could potentially spread from east to west Palearctic is likely underestimated, and the invasion deficit just east of the Urals may be higher than depicted in Fig 1.

Although our lists of naturalised species are not identical to the most up to date datasets of naturalised species [47–49] for birds and mammals this is unlikely to affect our results (S1 Text). Our initial list of plants, however, contained <20% of the naturalised plants recorded by [48], though the geographic pattern of naturalisations within realms, is broadly similar to that in [48] (S1 Text and S10 Fig). However, comparisons of the invasion deficit between plants and other taxa should not be made.

The importance of time since introduction, dispersal ability, and landscape connectivity suggest that with time, plants, birds, and mammals may overcome barriers and spread more widely within the hotspots we identify. For plants in particular, the time lags observed in potential spread hotspots in N. America and Australia (Figs 1 and S2) suggest that more recent introductions are highly likely to spread, though the same is not necessarily true in the European hotspot where the time lag was not significant. For many mammals, their high potential for spread in Eastern Europe and Australia (and potentially North America, though this effect was not significant) seems to have been slowed by fragmentation of suitable climate, which may be overcome with time. For birds, it is strange that more species have not spread widely in their primary hotspot of potential spread in North America, and the factors we investigated failed to explain why. In the Nearctic realm, bird introduction dates are somewhat older, the landscapes they encounter somewhat less fragmented, and their range of dispersal ability similar to other realms (S4 Fig). Further predictors of range filling will be required to help know whether the threat in this hotspot will be realised.

Given that the majority of introduced species have low impact [50], how should we interpret the threat in hotspots of potential spread? First, even if the effect of individual species is small, "invasional meltdown" is common when multiple introduced species co-occur and amplify each other's establishment and impact [51], and can be devastating [52, 53]. Although hard to predict, the more co-occurring introduced species in an area, the higher the likelihood of invasional meltdown. Second, species may not become problematic until after an initial

period of lag, spread, and population growth [6]. Invasion lags are clearly present in our study species, so even species that are currently unobtrusive could become problematic, and this is clearly more likely in hotspots of potential spread. Finally, the large invasion deficits suggest that climate-matching does not, by itself, predict invasive spread in the short term, and more attention must be paid to other factors that determine species' ranges.

## Materials and methods

### Collating naturalised species data and distributions

We identified birds, plants, and mammals that have established outside their native range following introduction by people. We included only species confirmed to have established and reproduced since 1770 on a mainland landmass outside their native continent, whose naturalised ranges do not rely exclusively on human activities such as irrigation or continued re-introduction, and for which we could clearly delineate their native range. For a full species list, see the Supporting information (S3–S5 Tables).

Plant species were drawn from those not listed as "Casual Alien," "unconfirmed naturalisation," "Contaminant," or "Native Weed" in Randall [54] and from the Global Invasive Species Information Network [55].

Bird species were compiled from known successful introduction events [41,56,57]. All migratory birds were removed from the list, due to difficulty defining a species' range and climatic niche. Migratory status was confirmed using Handbook of the Birds of the World [58].

Mammal species were compiled from Capellini and colleagues [59]. A search was made for additional mammal species from various sources, but in final analyses none of these species were included due to either a lack of data on the known naturalised range or because long-term establishment could not be confirmed.

For all species, we obtained occurrence data from GBIF (downloaded 31 August 2017) using R's dismo package [60]. For a record of all data sources, see the derived dataset at https://doi.org/doi.org/10.15468/dd.2zen56 [61]. Point data was then cleaned in several stages, firstly by removing points off-shore, then by removing points with low lat/lon precision (less than 10 arc-minute resolution), and then by removing points the exact centre of countries or states. Points were classified as either "native" or "naturalised" based upon sources listed in the Supporting information (S6 Table). Points that could not be validated using an independent state or national checklist as "native" or "naturalised" were discarded. Species that occupied fewer than 5 grid-cells after cleaning (at 10 arc-minute resolution) in their native or naturalised range were discarded. We mapped the potential and under-filled ranges for 65 mammal species, 114 bird species, and 616 plant species.

Although the process of filtering species to meet our criteria substantially reduced our species list, our initial sources of naturalised species are large and comprehensive. Therefore, we believe that our species lists are taxonomically broad, ecologically varied, and thus likely representative of the taxa introduced to each region. The 616 plant species in our list contain 391 genera and 116 families, the 114 bird species contain 28 families and 68 genera, and the 65 mammal species contain 50 genera and 24 families. In addition, while there is data bias towards some areas, notably Europe, North America, and Australia, there are examples of species in our database that are native to and naturalised in every major landmass on Earth except for Antarctica (S1 Fig).

We restricted predictions of each species' potential naturalised ranges to the biogeographic realm/s into which that species were introduced. We used a published set of biogeographic realms [62], but with an additional distinction between western and eastern Palearctic along an approximate line of the Ural Mountains (S11 Fig). This was done because species that

inhabited both western and eastern Palearctic were almost always native to one and naturalised in the other. The realms were created using multiple taxonomic groups, including birds, mammals, and plants, though the boundaries were similar to those created for each individual taxonomic group. For inter-taxonomic consistency, we used the same realms for every taxonomic group. Species that were found to be native and naturalised in the same biogeographic realm were removed from analysis due to the difficulty in exactly defining the native and naturalised ranges. If a species naturalised in multiple realms, each of the naturalised realms was examined separately.

## Modelling species potential naturalised ranges

We modelled potential ranges using 3 climate variables: mean temperature of coldest month, mean temperature of warmest month, and total annual precipitation. These represent the most universal parsimonious variables that influence species ranges [44]. Including a larger number of variables results in forecasts of smaller potential ranges, and less transferability than the parsimonious set of variables [44]. Gridded climate data were downloaded from World-Clim at 10 arc-minute resolution. Each grid-cell contained average climatic variables from 1970 to 2000 [63].

For each species, we extracted the climate conditions in the entire biogeographic realms to which they were native or naturalised. Climate variables were scaled so all variables had a mean of 0 and an SD of 1, total annual precipitation was logged prior to scaling. Any occurrences in climate conditions that were not found in both the native and naturalised biogeographic ranges (no-analogue climate) were removed from analysis. We used principal components analysis (PCA) to produce a gridded climate space of $100 \times 100$ cells on 2 axes [64]. Within this climate space, we applied a kernel smoothed density function to GBIF records in order to calculate species' occurrence densities in the climate conditions contained in each PCA grid-cell, which was then corrected by climate availability. A bivariate normal kernel was used, where the smoothing parameter was estimated using the ad hoc method, using the *kernelUD* function from the adehabitatHR package [65].

In order to measure each species' potential naturalised range, we identified the PCA grid-cells that contain climate the species occupies in its native realm, i.e., the grid-cells in which native density was above 0. Note that the kernel density function creates some infinitesimally small densities, so PCA grid-cells with densities less than one thousandth of the value of the grid-cell with the highest occurrence density were considered to have a value of 0. We then identified the geographic grid-cells in the naturalised realm to which these PCA grid-cells corresponded. We restricted predictions of species' potential naturalised ranges to the climate conditions present in both a species' native and naturalised realms (i.e., "analogue climate").

To measure the area of each species' potential naturalised range to which it has yet to spread, we constructed a naturalised range polygon for each species using its GBIF occurrence data and level 4 geographic administrative units from the TDWG scheme [66]. Within each administrative unit, we calculated the occupied area using a convex hull polygon around each species' occurrence data. We then aggregated all polygons for each species in each biogeographic realm. Any part of the species' potential naturalised range that was not contained within these range polygons was classed as unoccupied. Range filling for each species in each naturalised realm was calculated as the proportion of the climatically suitable area that was filled by the naturalised range polygons.

Defining the climate niche using all PCA grid-cells the species occupies in its native realm could include outlying distribution points in climate conditions a species can poorly tolerate,

inflating the species' climate niche. As a sensitivity test, we recalculated potential ranges using only climate that fell with the 70% most densely occupied climate in the native realm.

The approach we used has 2 principal advantages over more standard species distribution model (SDM) approaches. First, applying kernel smoothers to a PCA of native and naturalised regions accounts for climate availability, which has been shown to have major effects on estimates of species' potential naturalised ranges [64]. Second, the approach is akin to a "presence-only" SDM, which is more appropriate than a presence-absence SDM when modelling hundreds of species in multiple regions with highly varying recorder effort. If we were to have selected pseudo-absences to make presence-absence SDMs, in under-recorded areas many of these would have been false-absences, which would cause the SDM to underestimate species' potential naturalised ranges [67]. Given the potential effects of recorder effort on our results (S5 and S6 Figs), the degree of underestimation would vary strongly between species and regions, and would have been much more severe than with a presence-only approach.

## Validation of climate matching

We used cross-validation to evaluate how robust our climate matching method was, in particular whether we could have underestimated species' native climate niches. To this end, we randomly split the native occurrence data into a training set (80% of the data) and a validation set (the remaining 20%). This was repeated 5 times, each with a different 20% of the native data. For each validation dataset, we generated 20 sets of pseudo-absences, each with the same number of pseudo-absences as native occurrences. Pseudo-absences were randomly selected from any grid-cell in the native realm outside of administrative units known to have native occurrences. We then ran the PCA model on the occurrences in each training dataset as described above and calculated the true presence accuracy and TSS [68] against the corresponding validation presence and pseudo-absence datasets. True presence accuracy describes what proportion of the true presences in the validation dataset were correctly predicted as a presence, a value of 1 indicates prediction was 100% accurate and therefore that the estimated native climate niche is extremely robust. TSS is a measure of sensitivity and specificity and requires both presences and absences. TSS scores based on presence-only models are quite sensitive to the method of selecting pseudo-absences [69], but repeated sampling from the background data mitigates this effect. However, for native species with small ranges that are mostly likely not at climatic equilibrium, true presence accuracy is a better sensitivity measure.

## Sensitivity of invasion deficit estimates to recorder effort

We obtained published measures of recorder effort for plants [43, 70] and for birds and mammals, which compare the species that have GBIF records in a grid-cell with the species that are known to be in the region from surveys and expert knowledge. Meyer and colleagues [43] used these data to estimate the probability of detecting all known species in a given area. A value of zero indicates no recording effort and no species known to live there are detected, and a value of one indicates recording effort is sufficiently high that all species present will be recorded. Each 10-min grid-cell was assigned a detection probability by resampling from Meyer and colleagues [43] using a nearest neighbour method, which was necessary due to the different spatial resolutions of the 2 datasets. To calculate how numbers of native or naturalised species and potential for spread in each grid-cell might be altered once recording effort was compensated for, we multiplied the grid-cell value by the reciprocal of the detection probability. Detection probability was given a floor of 0.01% as otherwise the relative number of species once recording effort was accounted for could be hyper-inflated to unrealistic levels. It should be noted that Meyer and colleagues' estimates were based on the recording effort of native species, not

naturalised, but in the absence of quantitative data on the recording effort of naturalised species we thought it a sound assumption that the 2 would be closely correlated in most areas of the world. The corrected results do not necessarily represent a true prediction of the number of species, but highlight where uncertainty is highest and where large numbers of naturalised species are recorded despite poor recording effort. To our knowledge, this is the first time that a formal assessment of recorder effort has been applied to a global assessment of any aspect of biological invasions.

## Introduction history, species characteristics, and climate connectivity affecting range filling

**Introduction history.** The year of introduction is difficult to ascertain for most plants so the first confirmed record of occurrence in a realm was taken as the date of introduction, obtained from [71], the Atlas of Living Australia (ALA), Seebens and colleagues [72] and additional regional sources (see S7 Table). The year of introduction for birds was estimated using the first confirmed record from GAVIA [41] and Seebens and colleagues [71]. The year of introduction for mammals was estimated using the first confirmed record from Long [73] and Seebens and colleagues [71].

Previous studies have linked introduced species success to horticultural status and thus propagule pressure [74]. Therefore, whether a species was used in horticulture or not was extracted from Dave's Garden PlantFiles (http://davesgarden.com/guides/pf/, accessed 25 May 2018) and from the Plant Information Online database (https://plantinfo.umn.edu/, accessed 25 May 2018).

We investigated whether range filling depended on biogeographic realm of naturalisation, which could indicate anthropogenic factors influencing spread or biotic resistance [23,75,76].

**Species traits.** Rapid population growth likely increases range filling by allowing new populations to establish quickly and produce large numbers of propagules [21]. Traits associated with rapid population growth include time till sexual maturity [77] and seed/clutch size [21,59]. The age of a plant at first flowering and the seed number per flowering event were extracted from TRY [78]. For plants, we also investigated growth form obtained from TRY [78] and USDA Plants database. Plants were defined as either herbs, climbers, trees, shrubs, or ferns. For mammals, litter size, time till sexual maturity, and interbirth interval were taken from PanTHERIA [32]. For birds, clutch size and number of clutches per year were obtained from Myhrvold and colleagues [79].

Good competitive ability and adaptive capacity to novel environments could speed range expansion by allowing naturalised species to preempt resources from native species and invade novel niches. Corresponding traits include habitat generalism [16,17,21] and relative brain mass as an indicator of behavioural flexibility [16,80]. For both mammals and birds, habitat generalism was obtained for all species using the IUCN Habitat Classification Scheme (IUCN, 2023 [81]; accessed Nov 2018) and quantified as the number of general habitats as an integer and the number of sub-habitats as a decimal [21]. Information on brain residual size for mammals were taken from Sol and colleagues [80]. Data for brain size for birds was explored, but was not included as the data available would have resulted in a very limited sample size.

Dispersal ability has been linked to introduced species success, likely because it permits species to spread widely [22,36,77,82]. For many plant species, mean and maximum dispersal distance is frequently unknown, but dispersal distance can be estimated using a number of proxy life history traits [83]. Dispersal was estimated as a ranked category from 1 to 7, which correspond to increasing maximum dispersal distance on an approximately logarithmic scale [83]. Estimated dispersal distance varies depending on the species' dispersal mode,

plant height, habitat type, and taxonomic group [83]. We obtained the life history traits for calculating dispersal from the TRY database [78]. For birds, natal dispersal distance was estimated using diet, body mass, and wingspan [84]. Body mass and diet data were gathered from the EltonTraits database [85] and bird wingspan from del Hoyo and colleagues [58]. If only bird wing-length was available, bird wing-length was extrapolated to wingspan using the method in Garrard and colleagues [84]. For mammals, natal dispersal distance was estimated using body mass, home range size, and trophic level (for full method, see [86] from PanTHERIA [32]).

For all species, invasive status was determined by their description on GISIN (2015). Body mass was not used in the analysis as it covaried strongly with the estimated natal dispersal distance (Birds: Pearson's correlation R-squared = 0.54; Mammals: R-squared = 0.93).

**Spatial distribution of suitable areas.** We measured the fragmentation of climatically suitable areas in 2 different ways, the "contagion" of a landscape and the "clumpiness." Both were calculated in the FRAGSTATS program [87]. These 2 metrics measure different aspects of fragmentation and have different consequences for interpretation [88].

Contagion describes how dominant and aggregated climatically suitable grid-cells are over a landscape and how interspersed it is with non-suitable grid-cells [87]. Values for contagion range from 0 (climatically suitable grid-cells are completely fragmented and rare) to 100 (climatically suitable grid-cells are completely dominant across the landscape). Contagion is a good measure of fragmentation when comparing within similar areas, such as in the same geographic realm. However, contagion typically correlates with total available area, and as a result it is not a good measure of fragmentation when comparing across geographic realms of different sizes.

Clumpiness describes how spatially aggregated suitable grid-cells are over a landscape, after accounting for the overall abundance of suitable grid-cells [87]. Values for clumpiness range between −1 (maximally disaggregated), 0 (spatially randomly distributed), and 1 (maximally aggregated). Clumpiness considers only the number of cell-adjacencies as a proportion of the total number of cells, so does not typically correlate with total available area, and therefore can be used to compare fragmentation between realms [89].

**Local recording effort.** Ranges, and thus range filling, may be underestimated in areas of low recording effort. To investigate this, we used taxonomic coverage from the global sampling bias map of Meyer and colleagues [43]. For each species and realm, we calculated "local recording effort" as median taxonomic sampling coverage across all unoccupied grid-cells in the potential naturalised range.

For a summary list of all variables examined for each taxonomic group, see S8 Table.

## Statistical analysis of characteristics corresponding to range filling

We conducted analyses for species for which we could obtain the necessary data: 242 plant species (484 establishment events), 35 bird species (50 establishment events), and 22 mammal species (46 establishment events).

For each taxonomic group, the relationships between species' naturalised range filling and potential determinants were investigated using a hierarchical Bayesian model based on a beta distribution with a logit link. Realm, invasive status, growth form, horticultural status were treated as categorical, all other parameters were continuous. Body mass, home range size, and height were logged to improve linearity. All predictive parameters were centred on their mean and scaled by their standard deviation. We chose to use weakly informative priors throughout which favoured parameter estimates at or near zero (i.e., the null hypothesis that there is no relationship between predictor and response), but did not constrain the models from selecting

nonzero estimates [90]. For categorical and hierarchical effects, we used a weakly informative half-Cauchy distribution for the standard deviation among categorical levels. This reflects the null hypothesis that there is no difference between levels and biases the model towards conservative parameter estimates at or near 0, and therefore avoids overestimating the size of categorical and hierarchical effects. An example model with 1 continuous variable and 1 hierarchical effect is included in S9 Table.

Models were run using a Markov chain Monte Carlo (MCMC) method in JAGS through the R package "R2jags" with a burn-in of 10,000 samples and checked for convergence after a further 20,000 samples, which was extended if estimates did not converge. Models were evaluated using the DIC, widely applicable information criterion (WAIC), leave-one-out (LOO) evaluation [91], and Pearson's residual fit [92]. A pseudo-$R^2$ for each model was also calculated as a squared sample correlation between the mean linear predictors and the link-transformed response.

Due to the large number of parameters, we carried out an initial screening process to identify variables of interest. We first investigated the relationship between range filling in each taxonomic group and each predictive variable individually and ran univariate models with each variable in turn, with naturalised biogeographic realm included as a hierarchical effect. Posteriors were checked for a single unimodal peak, and predictive variables whose posterior estimate centred near 0 (meaning the value 0 lay between the 5th or 95th percentile parameter estimates) were not analysed further. We also tested for realm-specific effects, by identifying regions where parameter estimates did not overlap with other regions' (at 90% CI), and if variables showed strong trends in some realms but not others, they were retained in the final model.

In some cases, not all parameters identified by the univariate model process could be included in a multivariate simultaneously, due to strong co-variation in certain predictors. In these cases, we ran alternative multivariate models with either one of the predictors and compared results. In addition, models for birds and mammals with a full set of parameters (even when co-varying predictors were removed) frequently had issues with consistent convergence. If a model with a full set of parameters had issues with convergence or LOO, we trialled dropping parameters. In these cases, we would drop the least important variable in the full model to assist convergence, where the least important variable was defined as that with a mean parameter estimate nearest 0. If the resulting model still would not converge consistently, then another variable would be dropped. All of the variables dropped had mean parameter estimates at or near 0 (25% CI estimates all crossed 0), and no variable with obvious influence on the model was dropped in order to allow convergence. This process resulted in models that consistently produced stable, converged parameter estimates. Finally, we trialled several plausible, more parsimonious models by dropping less important variables to test if model fit could be further improved. All variables whose 95% CI estimates crossed 0 were trialled as candidates for dropping. The model was considered improved if there was a significant improvement in model fit ($\Delta$DIC >5).

All variable combinations tested are included in S10–S12 Tables. We considered the model with the lowest DIC value, and models with $\Delta$DIC $\leq$5 above this to be equally plausible, excluding any models with convergence issues. For mammals and birds this identified a single, best model. In the case of plants, 2 models were equally plausible, and their parameter estimates were extremely similar (see S13 Table). In the main paper, we report the most parsimonious of the 2 models, which excluded variables which had very little influence on the model and whose parameter estimates all strongly centred on 0.

In the multivariate models, a parameter was classed as having a significant effect when mean, 2.5th and 97.5th percentile estimates fell above or below 0. We confirmed models were

not mis-specified by checking that posterior parameter estimates were normal, as well as residual and LOO evaluation.

## Sensitivity analysis for low number of occurrences and outlying climate

Analysis of species range filling is potentially sensitive to bias when including species with extremely low record numbers. We set a threshold of a minimum of 5 naturalised grid-cells, which allows rarer species and species with small total potential ranges to be included in the analysis. However, species with very few records may reflect a lack of detection rather than a failure to fill their range, and therefore, species niches and ranges may not be accurately characterised. We accounted for this by including 2 forms of sensitivity analysis.

First, we tested for a correlation between range filling and the number of occupied naturalised grid-cells using a Bayesian mixed model (S9 Table). The number of grid-cells was logged for normality and treated as a continuous parameter, and region was also included as a hierarchical effect. If species with very low numbers of grid-cells show drastically different patterns in range filling to other species, then it suggests that the overall model is very sensitive to species with very small ranges, and a higher threshold should be considered.

Second, we re-performed the final model of range filling in plants, but with a higher cutoff threshold of 20 grid-cells. If the model is robust to species with very low numbers of grid-cells, the model output should return similar parameter estimates. Unfortunately, a similar analysis could not be completed for birds or mammals as the smaller sample size associated with a higher threshold prevented model convergence.

Code used to perform all analyses, derived data, and data necessary to replicate analyses are available at https://doi.org/10.5281/zenodo.8205905. Shapefiles of each species estimated native and naturalised ranges, rasters of niche filling and expansion, diagnostic and validation statistics are too large to place in a repository and so are available upon request from the authors.

## Supporting information

**S1 Text. Supplementary methods.**
(DOCX)

**S1 Table. Summary statistics of introduced species and range filling for all taxonomic groups across all biogeographic realms.** "Median range size" is the median range size (calculated with minimum convex hull polygon) of all established naturalised species (measured in 1,000 km$^2$). "Median potential range size" is the median area each species has available but has not colonised in each realm (measured in 1,000 km$^2$). For each species, the total occupied area is divided by the total available area to return a range filling proportion. "Median proportion of filling" is the median range filling across all species in the given realm.
(DOCX)

**S2 Table. Sensitivity analysis of correlates of range filling for plants.** This table presents results for the same model as in Table 1, but when only plants that have naturalised in >20 grid-cells were included. Estimates for parameters retained in the final model are given as the mean estimate of all posterior draws, with the 5% and 95% estimates as confidence intervals in parentheses. Parameter estimates are given as the linear slope of the logit link equation. When parameter estimates vary across realms, this is indicated by providing the names of the realms in which it varies (Aus = Australian, Nea = Nearctic, Neo = Neotropical). Model verification data are given for the final models, including sample size, DIC of the model, the effective

number of parameters (pD), and correlation of the linear predictor against the link transformed response given as a pseudo R-squared.
(DOCX)

**S3 Table. List of all plant species used in this study.**
(DOCX)

**S4 Table. List of all bird species used in this study.**
(DOCX)

**S5 Table. List of all mammal species used in this study.**
(DOCX)

**S6 Table. List of sources used to classify populations as native or naturalised.** Unless otherwise stated, plant sources were accessed June 2017, bird and mammal sources were accessed September 2017.
(DOCX)

**S7 Table. List of sources that provided either first introduction date or first observed record of introduced species.** Taxa indicate which taxonomic group the source covered. All sources accessed March 2018.
(DOCX)

**S8 Table. A summary of parameters trialled for plants, birds, and mammals.** All parameters in the "all variables trialled" list were included individually in a univariate Bayesian hierarchical model with region as the hierarchical effect. Parameters in bold in this column are variables that were significant (based on 90% CIs) either globally or in some at least some realms in the univariate model, and which were trialled in the multivariate model. Parameters in the "final variable list" are the parameters retained in the model for which results are described in the main manuscript.
(DOCX)

**S9 Table. An example Bayesian model based on a beta distribution with a logit link written in JAGs.** This model contains 1 continuous parameter (beta1) and 1 hierarchical effect (cat1). N is the total sample size, N_cat1 is the total number of levels of cat1, y is the response variable, x1 is the continuous predictor variable (scaled to have a mean of 0 and a standard deviation of 2).
(DOCX)

**S10 Table. A summary of all Bayesian hierarchical models trialled to correlate range filling with various traits and spatial features for plants.** Models 1 and 2 incorporate all variables that were important either at a global level or in some realms in univariate models (based on 90% CIs). Models were subsequently created as viable alternatives either to assist model convergence or to trial dropping unimportant variables (i.e., posterior estimates centred at or near 0). The final model used in the main manuscript is presented in bold. Alternative models that are equally viable are in italics.
(DOCX)

**S11 Table. A summary of all Bayesian hierarchical models trialled to correlate range filling with various traits and spatial features for birds.** Models 1 and 2 incorporate all variables that were important either at a global level or in some realms in univariate models (based on 90% CIs). Models were subsequently created as viable alternatives either to assist model convergence or to trial dropping unimportant variables. The final model used in the main

manuscript is presented in bold.
(DOCX)

**S12 Table. A summary of all Bayesian hierarchical models trialled to correlate range filling with various traits and spatial features for mammals.** Model 1 incorporates all variables that were important either at a global level or in some realms in univariate models (based on 90% CIs). Models were subsequently created as viable alternatives either to assist model convergence or to trial dropping unimportant variables. The final model used in the main manuscript is presented in bold.
(DOCX)

**S13 Table. Correlates of range filling for the 2 best, equally plausible models for plants.** Model numbers correspond to those in S10 Table. Estimates for parameters retained in the final model are given as the mean estimate of all posterior draws, with the 5% and 95% estimates as confidence intervals in parentheses. Parameter estimates are given as the linear slope of the logit link equation. When parameter estimates vary across realms, this is indicated by providing the names of the realms in which it varies (Aus = Australian, Nea = Nearctic, Neo = Neotropical). Model verification data are given for the final models, including sample size, DIC of the model, the effective number of parameters (pD), and correlation of the linear predictor against the link transformed response given as a pseudo R-squared. Note that a negative effect of recording effort means that more recorder effort in the potential naturalised range corresponded to lower range filling. Model 2 is presented in the main manuscript.
(DOCX)

**S1 Fig. Raw number of study species' native ranges (a, c, e) and naturalised ranges (b, d, f) that fall in each 10-min grid-cell.** Colours represent number of species. The data underlying this figure can be found in https://doi.org/10.5281/zenodo.8205905. Country and continent outlines were produced by the International Working Group on Taxonomic Databases for Plant Sciences (TDWG), specifically the WGSRPD Level 4 boundaries; data and usage notes can be found at (https://github.com/tdwg/wgsrpd).
(PNG)

**S2 Fig. Plotted parameter estimates of all variables kept in the final model to explain plant species range filling.** On the left are global trends for each parameter, on the right are the realm hierarchical effects. (a) Years since introduction, (b) age of first flowering event; (c) the estimated local sampling effort. A solid line signifies the estimate was consistently above or below 0 in >95% of simulations (and therefore judged as significant), a dashed line means it was not. The lighter shaded area shows the 95% probability density interval for the parameter estimate, and the darker shows the 50% interval. The data underlying this figure can be found in https://doi.org/10.5281/zenodo.8205905.
(PNG)

**S3 Fig. Plotted parameter estimates of all variables kept in the final mode to explain bird species range filling.** On the left are global trends for each parameter, on the right are the realm hierarchical effects. (a) Natal dispersal distance (km), (b) fragmentation of suitable climate (contagion), (c) years since introduction. A solid line signifies the estimate was consistently above or below 0 in >95% of simulations (and therefore judged as significant), a dashed line means it was not. The lighter shaded area shows the 95% probability density interval for the parameter estimate, and the darker shows the 50% interval. The data underlying this figure can be found in https://doi.org/10.5281/zenodo.8205905.
(PNG)

**S4 Fig. Plotted parameter estimates of all variables kept in the final model to explain mammal species range filling.** On the left are global trends for each parameter, on the right are the realm hierarchical effects. (a) Fragmentation of suitable climate (contagion), (b) dispersal distance (logged km). A solid line signifies the estimate was consistently above or below 0 in >95% of simulations (and therefore judged as significant), a dashed line means it was not. The lighter shaded area shows the 95% probability density interval for the parameter estimate, and the darker shows the 50% interval. The data underlying this figure can be found in https://doi.org/10.5281/zenodo.8205905.
(PNG)

**S5 Fig. Number of study species' native ranges (a, c, e) and naturalised ranges (b, d, f) that fall in each 10-min grid-cell, after adjustment for recording effort (see Methods and Meyer and colleagues for a full description).** Colours represent the adjusted relative number of species. The data underlying this figure can be found in https://doi.org/10.5281/zenodo.8205905. Country and continent outlines were produced by the International Working Group on Taxonomic Databases for Plant Sciences (TDWG), specifically the WGSRPD Level 4 boundaries; data and usage notes can be found at (https://github.com/tdwg/wgsrpd).
(PNG)

**S6 Fig. Relative threat from the spread of regionally naturalised species globally after adjustment for recording effort for (a) birds, (b) mammals, and (c) plants.** The number of species that could spread to each 10-min grid-cell is calculated in the same way as in Fig 1). This number was then multiplied by a measure of recording effort (proportion of known species per grid-cell that are actually reported in GBIF data, see Methods and Meyer and colleagues for a full description) to compensate for potential over- or under-recording of species. The data underlying this figure can be found in https://doi.org/10.5281/zenodo.8205905. Country and continent outlines were produced by the International Working Group on Taxonomic Databases for Plant Sciences (TDWG), specifically the WGSRPD Level 4 boundaries; data and usage notes can be found at (https://github.com/tdwg/wgsrpd).
(PNG)

**S7 Fig. Correlation of number of naturalised grid-cells with proportion of range filling for plants (a), birds (b), and mammals (c).** A solid line signifies the estimate was consistently above or below 0 in >95% of simulations (and therefore judged as significant), a dashed line means it was not. The lighter shaded area shows the 95% probability density interval for the parameter estimate, and the darker shows the 50% interval. Point colour represents region, but as parameter estimates did not vary between region, only the global regression line is shown. The data underlying this figure can be found in https://doi.org/10.5281/zenodo.8205905.
(PNG)

**S8 Fig. Correlation of number of naturalised grid-cells with proportion of niche filling for plants (a), birds (b), and mammals (c).** A solid line signifies the estimate was consistently above or below 0 in >95% of simulations (and therefore judged as significant), a dashed line means it was not. The lighter shaded area shows the 95% probability density interval for the parameter estimate, and the darker shows the 50% interval. The data underlying this figure can be found in https://doi.org/10.5281/zenodo.8205905.
(PNG)

**S9 Fig. Relative threat from the spread of regionally naturalised species globally using only climate that fell with the 70% most densely occupied climate in the native realm for a+b) plants, c+d) birds, and e+f) plants.** Figures on the left depict the number of species that could

spread to that area and are already naturalised within that realm. Figures on the right depict the discrepancy between threat metrics for 100% (Fig 1) and 70% niche overlap maps. Positive values mean that the number of species that are calculated to spread using 100% of occupied native climate is greater than when using 70% of occupied native climate. The data underlying this figure can be found in https://doi.org/10.5281/zenodo.8205905. Country and continent outlines were produced by the International Working Group on Taxonomic Databases for Plant Sciences (TDWG), specifically the WGSRPD Level 4 boundaries; data and usage notes can be found at (https://github.com/tdwg/wgsrpd).
(PNG)

**S10 Fig. The proportion of species that fall in different geographic regions compared between our study and published databases for plants (GLONAF), birds (GAVIA), and mammals (DAMA).** Note that the geographic regions are specific to each published database, not to our study. Also note that proportions for published databases and for our study sum to >1, as some species have naturalised in multiple regions. The data underlying this figure can be found in https://doi.org/10.5281/zenodo.8205905.
(PNG)

**S11 Fig. Biogeographic realms used in this paper.** Realms were defined by Holt and colleagues, with the addition of a line between the western and eastern Palearctic along the Ural Mountains. The data for this figure is not available from the authors of this study, but can be attained from Holt and colleagues. Credit: Journal Science/AAAS.
(PNG)

## Acknowledgments

We would like to thank C. Meyer for providing data on global recording effort, Wayne Dawson and Chris Kaiser-Bunbury for comments on an earlier draft. Some of the text in this manuscript already appears in a thesis by Dr. Hakkinen [93].

## Author Contributions

**Conceptualization:** Henry Häkkinen, Regan Early.

**Data curation:** Henry Häkkinen.

**Formal analysis:** Henry Häkkinen.

**Funding acquisition:** Regan Early.

**Investigation:** Henry Häkkinen, Regan Early.

**Methodology:** Henry Häkkinen.

**Project administration:** Regan Early.

**Resources:** Regan Early.

**Supervision:** Dave Hodgson, Regan Early.

**Visualization:** Henry Häkkinen.

**Writing – original draft:** Henry Häkkinen.

**Writing – review & editing:** Dave Hodgson, Regan Early.

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
