## [Editor Report · Decision Letter 0]

18 May 2023

Dear Dr Hakkinen, 

Thank you for submitting your revised manuscript entitled "Global terrestrial invasion deficits and what causes them." for consideration as a Research Article by PLOS Biology.

Your manuscript has now been evaluated by the PLOS Biology editorial staff. I'm still waiting to hear back from the Academic Editor, but in the interim, I'm asking you to complete your submission on the assumption that the AE will agree to send your submission out for re-review.

IMPORTANT: Please can you change the article type to "Short Reports" when you upload your additional metadata (see next paragraph)?

Before we can send your manuscript to reviewers, we need you to complete your submission by providing the metadata that is required for full assessment. To this end, please login to Editorial Manager where you will find the paper in the 'Submissions Needing Revisions' folder on your homepage. Please click 'Revise Submission' from the Action Links and complete all additional questions in the submission questionnaire.

Once your full submission is complete, your paper will undergo a series of checks in preparation for peer review. After your manuscript has passed the checks it will be sent out for review. To provide the metadata for your submission, please Login to Editorial Manager (https://www.editorialmanager.com/pbiology) within two working days, i.e. by May 23 2023 11:59PM.

Kind regards,

Roli Roberts

Roland Roberts, PhD

Senior Editor

PLOS Biology

rroberts@plos.org

---

## [Decision Letter · Decision Letter 1]

4 Jul 2023

Dear Dr Early,

Thank you for your patience while we considered your revised manuscript "Global terrestrial invasion deficits and what causes them." for consideration as a Short Report at PLOS Biology. Your revised study has now been evaluated by the PLOS Biology editors, the Academic Editor and one of the original reviewers.

In light of the reviews, which you will find at the end of this email, we are pleased to offer you the opportunity to address the remaining points from the reviewers in a revision that we anticipate should not take you very long. In addition, the Academic Editor has provided some guidance which I've pasted into the foot of the email. We will then assess your revised manuscript and your response to the reviewers' comments with our Academic Editor aiming to avoid further rounds of peer-review, although might need to consult with the reviewer, depending on the nature of the revisions.

IMPORTANT:

a) We're finding the current Title somewhat opaque, and wonder if it might be better to include more specific findings; as you say in the Abstract, "Prediction of terrestrial regions to which 833 naturalized plants, birds and mammals are most imminently likely to spread" - could you fashion a Title more along those lines? (though I do note that you also explore the underlying drivers of this)

b) Please address the concerns raised by reviewer #1 and the Academic Editor.

c) Please address my Data Policy requests below; specifically, we need you to supply the numerical values underlying Figs 1ABC (note that panels are in the wrong order!), 2ABC, S1ABCDEF, S2ABC, S3ABC, S4AB, S5ABCDEF, S6ABC, S7ABC, S8ABC, S9ABCDEF, S10ABCDEF, S11ABC, S13, either as a supplementary data file or as a permanent DOI’d deposition. I note that the raw data used in your study belong to third party sources; such data are exempt from our policy as long as readers can obtain them in a similar manner; however, we will need the processed numerical values that directly underlie the Figure panels.

d) Please cite the location of the data clearly in all relevant main and supplementary Figure legends, e.g. “The data underlying this Figure can be found in S1 Data” or “The data underlying this Figure can be found in https://doi.org/10.5281/zenodo.XXXXX”

**IMPORTANT - SUBMITTING YOUR REVISION**

*Resubmission Checklist*

*Published Peer Review*

*PLOS Data Policy*

*Blot and Gel Data Policy*

Sincerely,

Roli Roberts

Roland Roberts, PhD

Senior Editor

PLOS Biology

rroberts@plos.org

REVIEWERS' COMMENTS:

Reviewer #1:

PBIOL-D-23-01216.R1

The revision has improved the paper a lot and I also like the new focus. The authors appropriately justify their methods (but see below). My comments on the revision are generally minor, if the statistical approach chosen by the authors holds up to scrutiny and results do not change if a different approach would be chosen. 

I'd like the authors to consider three general issues in their revision: First, the description of which parameters have been tested in each taxon (plants, birds, mammals) was sometimes unclear and should be carefully revised. I noticed some obvious errors below in the detailed comments, but the entire section would profit from a careful revision. Second, the model selection strategy (step-wise procedures; reliance on one "best" model) chosen by the authors has been criticised a lot in the statistical literature. Although in my experience results from such analyses are often very similar to those from recommended "superior" analyses (e.g. comparison of all plausible models; inference considering multiple models), I think the paper would be strengthened if the authors would justify their choice of statistical procedure. Otherwise, a switch to less contested methods might be more appropriate, but it is difficult to estimate how much this would change the outcomes. Finally, I find it unfortunate that the authors do not make the data available by claiming not to be the property holders of the data. Thus, it will be close to impossible to re-run the analyses to confirm the integrity of the results. 

Detailed comments:

Table 1: please harmonise whether WAIC (legend) or DIC (table) was used. 

l. 198: maybe give references for the effect of these traits in affecting invasion (e.g. some of these traits have been shown to affect establishment success in both birds and mammals)

l. 204: interesting hypothesis; is this independently supported in the literature?

l. 224ff: This argument would also apply to other taxa, but here you seem to have found evidence mainly for birds. Could you discuss this?

l. 359: add reference

l. 368: delete "they"

l. 433: ALA = Atlas of Living Australia?

l. 452-8: It is unclear to me if this applies to all taxa (plants, birds, mammals) or only to some. Please specify. E.g. the reference for brain size only applies to mammals, but brain size information is also available for birds (reference 55 in the paper)

l. 469: for mammals? 

l. 482: are dominant

l. 487: "their overall abundance"? please clarify to what this refers

l. 521ff: The model selection process follows a step-wise strategy, which I guess is at least partly due to the computational complexity of conducting a comparison of models with all possible combinations of factors, or a comparison of all "plausible" models (i.e. those that could be defended without knowing the results). Although in my experience step-wise model selection and comparison of all possible models often yields very similar results, the step-wise selection process has been widely criticised in the literature (see references below) and I think that some justification for this approach would convince readers of the robustness of the findings and leave authors less exposed to criticism. Similarly, relying on one "best-fitting" model is usually considered as inferior to drawing inference over all credible models (e.g. by model averaging); also this choice would profit from more justification.

l. 529: delete (pD)?

Figure 1: please arrange letters a,b,c from top to bottom

Figure 1: there is a strong divide in the number of species that could expand their alien ranges across the Ural Mountains in all 3 taxa and I wonder if this might be an artefact e.g. due to the choice of separating east and west Palaearctic or recording bias (although the authors seem to rule out the latter). I didn't see in the map any other transition of similar strength, thus I'd like the authors to discuss this and particularly why this is the same in all taxa. 

Anderson, D. R., & Burnham, K. P. (2002). Avoiding pitfalls when using information-theoretic methods. The Journal of wildlife management, 66, 912-918.

Burnham, K. P., Anderson, D. R., & Huyvaert, K. P. (2011). AIC model selection and multimodel inference in behavioral ecology: some background, observations, and comparisons. Behavioral ecology and Sociobiology, 65, 23-35.

Whittingham, M. J., Stephens, P. A., Bradbury, R. B., & Freckleton, R. P. (2006). Why do we still use stepwise modelling in ecology and behaviour?. Journal of animal ecology, 75(5), 1182-1189.

COMMENTS FROM THE ACADEMIC EDITOR [lightly edited]:

The revision done in this version has been exhaustive and the authors did a good job. The three issues pointed out by reviewer #1 require further revision, however.

A small table in Suppl Mat describing in detail the parameters used in each group could be very useful and clarify very much the overall approach.

The authors need to carefully consider the model selection approach. Stepwise procedures have been critiqued recently and most recent approaches to model selection are done with modern model-combination techniques (e.g., multi-model averaging, etc.). Please see, e.g.: Whittingham et al. 2006 J Animal Ecol. https://doi.org/10.1111/j.1365-2656.2006.01141.x

In brief, stepwise selection procedures have three main weaknesses: bias in parameter estimation, inconsistencies among model selection algorithms, and an inappropriate focus or reliance on a single "best model". I'd recommend the authors to carefully consider an improved approach to their model selection strategy.

I also agree with the issue about data availability. This is a must nowadays, especially for this type of compilation paper. PLOS Biology has detailed instructions and requirements about data availability.

---

## [Editor Report · Decision Letter 2]

4 Oct 2023

Dear Dr Early,

Thank you for the submission of your revised Short Report "Global terrestrial invasions: where might naturalised birds, mammals, and plants spread next and what is stopping them?" for publication in PLOS Biology. On behalf of my colleagues and the Academic Editor, Pedro Jordano, I'm pleased to say that we can in principle accept your manuscript for publication, provided you address any remaining formatting and reporting issues. These will be detailed in an email you should receive within 2-3 business days from our colleagues in the journal operations team; no action is required from you until then. Please note that we will not be able to formally accept your manuscript and schedule it for publication until you have completed any requested changes.

IMPORTANT: We’d like to change the Title to make it more explicit and appealing for our broad readership, and to reduce the amount of punctuation. Here are the suggestions from my colleagues:

a) "Global terrestrial invasions: where naturalized birds, mammals and plants might spread next and what is stopping them" (this is most like the current one, and simply removes the rhetorical question)

b) "Prediction of where invasive species of birds, mammals and plants will spread next at a global scale" (this lacks the bit about drivers)

c) "Global prediction of where 833 invading species of birds, mammals and plants are likely to spread next" (this lacks the bit about drivers)

d) "Global prediction of where invasive species of birds, mammals and plants are likely to spread next and the factors affecting this process" (this is probably our favourite)

Ideally we'd like you to change your Title to option 'd', but if you're unhappy with all of options 'b', 'c' and 'd', please feel free to discuss an alternative with me via email (rroberts@plos.org). I've alerted my colleagues to expect a Title change.

Sincerely, 

Roli Roberts

Senior Editor

PLOS Biology

rroberts@plos.org